# Toll-like Receptor 3 in the Hybrid Yellow Catfish (*Pelteobagrus fulvidraco ♀ × P. vachelli ♂*): Protein Structure, Evolution and Immune Response to Exogenous *Aeromonas hydrophila* and Poly (I:C) Stimuli

**DOI:** 10.3390/ani13020288

**Published:** 2023-01-14

**Authors:** Shengtao Guo, Mengsha Zeng, Wenxue Gao, Fan Li, Xiuying Wei, Qiong Shi, Zhengyong Wen, Zhaobin Song

**Affiliations:** 1Key Laboratory of Bio-Resources and Eco-Environment of Ministry of Education, College of Life Sciences, Sichuan University, Chengdu 610065, China; 2Key Laboratory of Sichuan Province for Fishes Conservation and Utilization in the Upper Reaches of the Yangtze River, Neijiang Normal University, Neijiang 641100, China; 3State Key Laboratory of Protein and Plant Gene Research, School of Life Sciences, Peking University, Beijing 100871, China; 4Shenzhen Key Lab of Marine Genomics, Guangdong Provincial Key Lab of Molecular Breeding in Marine Economic Animals, BGI Academy of Marine Sciences, BGI Marine, BGI, Shenzhen 518083, China

**Keywords:** hybrid yellow catfish, Toll-like receptor 3, *Aeromonas hydrophila*, Poly (I:C), immunity

## Abstract

**Simple Summary:**

Yellow catfish is an economically important fish that lives in the middle and upper reaches of the Yangtze River and the Pearl River Basin in China. Due to the intensive farming and overuse of drugs, viral and bacterial diseases have emerged and have caused serious problems in domestic fisheries during recent years and thus threaten the production of aquaculture. It has been proved that exploring disease-resistant varieties is the key way to raising the aquaculture production of yellow catfish. Previous research showed that TLR3 is involved in the antiviral response in mammals. However, the molecular mechanism of immune response against viruses in teleost fish remains unclear. In this study, the gene structure and evolutionary conservation of TLR3 in teleost fish, as well as the tissue distribution in hybrid yellow catfish and its response to Aeromonas hydrophila and poly (I:C) at different time points, were systematically investigated. Our results showed that TLR3 played a significant role in hybrid yellow catfish response to viral infection. Our study provides theoretical basis for further research on the molecular mechanism of antivirus response in fish and the cultivation of hybrid yellow catfish with high resistance.

**Abstract:**

As a major mediator of cellular response to viral infection in mammals, Toll-like receptor 3 (TLR3) was proved to respond to double-stranded RNA (dsRNA). However, the molecular mechanism by which TLR3 functions in the viral infection response in teleosts remains to be investigated. In this study, the *Toll-like receptor 3* gene of the hybrid yellow catfish was identified and characterized by comparative genomics. Furthermore, multiple sequence alignment, genomic synteny and phylogenetic analysis suggested that the homologous TLR3 genes were unique to teleosts. Gene structure analysis showed that five exons and four introns were common components of TLR3s in the 12 examined species, and interestingly the third exon in teleosts was the same length of 194 bp. Genomic synteny analysis indicated that TLR3s were highly conserved in various teleosts, with similar organizations of gene arrangement. De novo predictions showed that TLR3s were horseshoe-shaped in multiple taxa except for avian (with a round-shaped structure). Phylogenetic topology showed that the evolution of TLR3 was consistent with the evolution of the studied species. Selection analysis showed that the evolution rates of TLR3 proteins were usually higher than those of TLR3-TIR domains, indicating that the latter were more conserved. Tissue distribution analysis showed that TLR3s were widely distributed in the 12 tested tissues, with the highest transcriptions in liver and intestine. In addition, the transcription levels of TLR3 were significantly increased in immune-related tissues after infection of exogenous *Aeromonas hydrophila* and poly (I:C). Molecular docking showed that TLR3 in teleosts forms a complex with poly (I:C). In summary, our present results suggest that TLR3 is a pattern recognition receptor (PRR) gene in the immune response to pathogen infections in hybrid yellow catfish.

## 1. Introduction

Toll-like receptors (TLRs) have been validated to be the major sensors in signal transduction of innate immunity [1]. There are 13 members in the TLR family of mammals [2]. They are subdivided into cell-surface TLRs (including TLR1, 2, 4, 5, 6 and 10) and intracellular TLRs (including TLR3, 7, 8, 9, 11, 12 and 13) based on their cellular localization [3,4]. Previous studies reported that lipids, lipoproteins and proteins are recognized by cell-surface TLRs in the microbial membrane [3]. Nucleic acids in bacteria and viruses are recognized by the intracellular TLRs in both healthy and disease conditions [5]. It has been found that viral double-stranded RNA (dsRNA), small interfering RNA and self RNA from damaged cells were recognized by TRL3 [6,7]; in this process, TRL3 triggers the inflammatory response and therefore weakens the spreading of most bacteria and viruses [8].

In mammals, TLR3 recognizes pathogen-derived double-stranded RNAs (dsRNAs; [9]). In teleosts, however, analog structures such as poly (I:C) in dsRNAs can also mediate antiviral immune responses [10,11]. As we know, ray-finned fishes represent a transition point in the phylogenetic spectrum between invertebrates and mammals, among which the former rely solely on innate immunity while the latter depend heavily on adaptive immunity [12]. The main mechanisms of the innate immune response in teleost fishes are thought to be similar to those in mammals [13]. As in mammals, viral infection in teleost fishes may mediate the expression of many host genes [14]. Previous studies showed that injection of exogenous poly (I:C) into common carps (*Cyprinus carpio*) resulted in upregulation of the antiviral *Mx* gene in spleen, liver and kidney [15]. Meanwhile, the transcriptions of IFN regulatory factor 3 (*irf3*; playing an indispensable role in the signal transduction of TLR3) in orange-spotted grouper (*Epinephelus coioides*) were upregulated, with the stimulation of poly (I:C) in spleen [16]. Previous studies have reported the function of TLR3 in the immune response in teleosts [17]. For example, TLR3 in Japanese sea perch was found highly expressed in the immune tissues and upregulated significantly after bacterial infection [18]. Simultaneously, findings showed that TLR3 in Japanese sea perch functions in head kidney, spleen and liver in a time-dependent manner [18]. TLR3 in the immune response of golden pompano may act as a pattern recognition receptor (PRR) transmitting signals to downstream pathways against the pathogen infection [19]. Furthermore, TLR3 has been found to participate in the immune process against bacterial infection in hybrid yellow catfish [20]. Upregulation of TLR3 transcripts in spleen tissue following polyinosinic–polycytidylic acid (poly I:C) stimulation in yellow catfish suggested that TLR3 plays a key role in recognizing dsRNA and initiating immune responses against dsRNA viruses as well [20,21]. Consequently, immune responses, mediated by TLR3 in teleosts, may be crucial in viral infections as in mammals. However, the molecular mechanisms of fish TLR3s’ activation on signaling and innate immunity in response to exogenous dsRNA viruses are still not well established.

To understand the changes in proteins under macromolecular crowding conditions in cells, it is essential to analyze their spatial structures [22]. AlphaFold has been proved to predict protein structures with high accuracy, and this bioinformatics tool integrates biological as well as physical knowledge of protein structures [23]. Homologous genes among different species can be analyzed by a comparative genomic method [24], and the publication of a chromosomal-level genome assembly of yellow catfish provides a valuable genetic resource for any genome-based collinearity analysis [25]. In order to explore the selection pressure on coding genes, the classic approach is to calculate the ratio of the nonsynonymous to synonymous substitution (dN/dS) rates [26,27]. Natural selection analysis and phylogenetic relationship construction of coding genes can uncover the mechanisms of molecular evolution [27,28].

Multiple sequence alignment, gene structure, linearity analysis, protein structure, phylogenetic analysis and selection analysis were employed in the present study. The TLR3 gene in yellow catfish was identified through a comparative genomic approach; simultaneously, the TLR3 conservation in teleosts, their protein structures, evolutionary relationship and natural selection in multiple biological groups were examined. Finally, tissue distribution and transcriptional changes in hybrid yellow catfish TLR3 in multiple immune tissues after bacterial and viral stimulation were analyzed.

## 2. Materials and Methods

### 2.1. Gene Identification and Multiple Sequence Alignment

The genome data of yellow catfish were downloaded from NCBI (the GenBank assembly accession number is GCA_022655615.1). BLAST search based on the similarity of amino acid sequences was conducted to identify the TLR3 gene in the genome as well as transcriptome data (SRX16687130) of yellow catfish from previous research [29]. ClustalW (http://www.ebi.ac.uk/clustalw/, accessed on 20 July 2022) and the Molecular Evolution Genetics Analysis (MEGA) X software were applied to perform multiple alignment of TLR3 amino acid sequences from ten representative teleost species. The secondary structure of TLR3 was predicted using an online tool (https://npsa-prabi.ibcp.fr, accessed on 20 July 2022).

### 2.2. Data Processing and Bioinformatics Analysis

Spatial structures of TLR3s from different taxa were predicted using the DeepMind AlphaFold2 (https://github.com/deepmind/alphafold, accessed on 20 July 2022; [23]). Subsequently, gene synteny of TLR3 based on a series of representative teleost genomes was investigated through the comparative genomic survey in previous research [30]. Ten teleost species (Pelteobagrus fulvidraco, Cheilinus undulatus, Gambusia affinis, Oryzias latipes, Sparus aurata, Triplophysa tibetana, Chelmon rostratus, Acanthchromis polyacanthus, Betta splendens and Echeneis naucrates) were selected to predict the homologous regions in different genomes. Genomes and corresponding annotation files of ten fish species were downloaded from NCBI. Protein sequences of these representative TLR3s in fishes were identified by gene annotation and confirmed with functional motifs and domains. Genomic loci of TLR3 genes in yellow catfish as well as other fishes were identified by using the comparative genomic synteny method, and the corresponding 5–10 upstream and downstream genes were selected for linear analysis to determine the location of TLR3.

### 2.3. Phylogenetic and Natural Selection Analysis of TLR3 Genes

The phylogenetic trees, based on the aligned protein datasets with the maximum-likelihood method, were constructed using MEGA X [31]. Models were calculated and evaluated using MrmodelTest 2.0 and ProtTest 2.4 [32], and the JTT+G model was selected as the best for further studies. Constructed phylogenetic trees were optimized using the online tool iTOL (https://itol.embl.de/, accessed on 11 August 2022). A nonparametric guided analysis was performed to investigate the robustness of the tree topology with 1000 resampling replicates. See more details of the complete protein sequences in Appendix A. Domain structures of TLR3s were predicted using the Simple Modular Architecture Research Tool according to a previous study [33].

The genomic data of six vertebrate classes (including Mammalia, Amphibia, Aves, Chondrichthyes, Reptilia and Osteichthyes) and one invertebrate class (Arthropoda) were downloaded from NCBI to explore the evolutionary history of TLR3. HMMER was applied to BLAST against the pfam database and detect the TIR domain architectures of all TLR candidates [34]. Full-length TLR proteins and the TIR domains of each TLR candidate were selected for subsequent analyses. The alignment of the coding sequence of each *TLR* gene was performed using the MUSCLE (align codons) software [35]. Meanwhile, the PAML v4.7 package implemented with the codeml method [36] was employed in the present study to estimate dN/dS values [37,38].

### 2.4. Fish Sampling and Immune Challenges

A group of 150 hybrid yellow catfish (body weight of 37 ± 0.5 g) were purchased from a local aquaculture breeding base in the city of Neijiang, Sichuan Province, China. They were shipped to the aquaculture base at Neijiang Normal University for practical infection experiments.

These fishes were cultured in 200 L tanks (labeled as pond 1, pond 2 and pond 3, respectively) at 27 °C. They were fed with commercial feed (Tongwei Biotechnology Co. Ltd., Chengdu, China; with 40% protein) at 18:00 daily. After acclimatization for two weeks, three healthy fishes were selected for the tissue distribution study. During the two weeks of temporary breeding, fishes with poor mobility were discarded to ensure that fishes used for subsequent experiments were healthy. Furthermore, the active and shiny fish in this batch were selected for follow-up experiments. Twelve tissue samples, including caudal fin, barbel, gallbladder, intestine, liver, stomach, heart, gills, spleen, kidney, skin and brain, were collected from each fish to explore the distribution of TLR3 in corresponding tissues of the hybrid yellow catfish.

*A. hydrophila* was used as a bacterial pathogen [39], in vivo grade poly (I:C) was purchased from Invivogen (Invivogen, San Diego, CA, USA) and the virus-like challenge was performed using the poly (I:C) in this study. Transcription levels of the TLR3 gene in dissected tissues were investigated after the intraperitoneal injections of *A. hydrophila* and poly (I:C). Fishes in pond 2 were intraperitoneally injected with 100 μL (2.8 × 10^9^ CFU/mL) of *A. hydrophila* suspension to perform the bacterial challenge [40]. Fishes in pond 1, injected with an equal volume of sterile phosphate buffer saline (PBS), were set as the controls. Immune-related tissues such as spleen and kidney of three fishes were sampled at 0, 3, 6, 12, 24 and 48 h post-injection (hpi) to explore the expression changes in TLR3 after *Aeromonas hydrophila* and poly (I:C) challenge. Fishes in pond 3 were injected with 50 μL (1 mg/mL) of poly (I:C) (InvivoGen, Hong Kong, China) to perform the virus challenge. At each time point of the treatment, immune-related tissues (spleen and kidney) from three individuals were collected. Tissues were frozen in liquid nitrogen and stored at −80 °C until use. All animal experiments in this study were approved by the Ethics Committee of the College of Life Sciences, Sichuan University.

### 2.5. RNA Extraction, Reverse Transcription and Quantitative Real-Time PCR

Total RNA from 12 tissues was extracted to study the tissue distribution of TLR3, and then RNA was extracted at six time points (at 0, 3, 6, 12, 24 and 48 h post-injection) from spleen and kidney to study the changes in TLR3 expression before and after infection with pathogens. Each tissue and each time point had three biological replications. Extraction of total RNA and synthesis of the first-strand cDNAs were performed according to the manufacturer instructions (Tiangen, Beijing, China). One microgram of total RNA was used to generate the first-strand cDNA using the cDNA Synthesis Kit (TIANGEN, Beijing, China) according to the manufacturer’s instructions. RNase-free DNase (TIANGEN Biotech, China) was used to remove the genomic DNA according to the manufacturer’s instructions. Subsequently, quantitative RT-PCRs were conducted to measure the transcriptional levels of TLR3 on a Bio-Rad T100 Thermal Cycler amplifier (Bio-Rad, Hercules, CA, USA). The total volume of each PCR was 20 μL, including 10 μL of SYBR Premix Ex Taq (Tiangen Biotech), 40 ng of cDNA, 0.2 mM of each sense and antisense primer and 7 μL of double distilled H₂O. The reactions were carried out in a three-step procedure as follows: initially at 95 °C for 1 min, followed by 40 cycles of 95 °C for 5 s, 60 °C for 20 s and 72 °C for 20 s. The final PCR products generated in the quantitative PCRs were verified as a melting curve (with a single peak) for the target gene. Relative expression levels of hybrid yellow catfish TLR3 were obtained as reported before (29), and presented as mean ± SEM (*n* = 3). *β-actin* was used as the internal control gene. Details of the primer pairs used in this section are provided in Table 1.

### 2.6. Interaction of dsRNA Virus and Poly (I:C) with TLR3 in Yellow Catfish

The 46 bp dsRNA (AUUCUGCGGAUUAUUUGGCAAAGGAAGCAUUGACACAUGCGCCAAU) virus which forms a complex with the TLR3 ectodomain (PDB ID: 3CIY) in mice [41] and poly (I:C) were used for molecular docking with yellow catfish TLR3, loach TLR3 and mouse TLR3, respectively. Predictions of the secondary structure were performed using RNA2 (http://www.genebee.msu.su/genebee.html, accessed on 11 August 2022) and the folds of those well-preserved dsRNA regions were analyzed as well. Finally, the docking was completed using molecular docking software (HADDOCK) with reference to previous research [41,42].

### 2.7. Data Statistical Analysis

In this study, all data are presented as mean ± SEM. Statistical evaluation was performed using the repeated-measures one-way ANOVA in GraphPad Prism 9 [43], and the *t*-test as well as Duncan’s multiple range test were conducted. When *p* < 0.05, any difference was considered as significant at the statistical level.

## 3. Results

### 3.1. Multiple Sequence Alignment of TLR3s from Different Fishes

The full-length cDNA sequences of 10 TLR3s were downloaded from the NCBI database, and the detailed accession numbers and sequence descriptions are provided in Appendix A. cDNA of the 10 *TLR3s* range from 2739 to 5578 bp, among which 2712–2874 bp open reading frames were predicted to encode proteins with 903–957 amino acids. In order to investigate the structural properties and function of TLR3 proteins in various fishes, we performed multiple sequence alignments. Our analysis of amino acid sequence homology in different fishes indicated that TLR3 in yellow catfish exhibited somewhat lower levels of sequence identity and similarity compared to most of the other teleost TLR3s previously reported (Figure 1). According to similarity analyses in this study, TLR3 yellow catfish is closely related to TLR3 in channel catfish (*Ictalurus punctatus*) with similarity of 90.1% (Table 2). Multiple sequence analysis found that TLR3 in yellow catfish is poorly conserved among other teleosts such as spiny chromis (*Acanthochromis polyacanthus*), humphead wrasse (*Cheilinus undulatus*), European bass (*Dicentrarchus labrax*), Indo-Pacific tarpon (*Megalops cyprinoides*), Nile tilapia (*Oreochromis niloticus*), brown trout (*Salmo trutta*) and mandarin fish (*Siniperca chuatsi*) (Figure 1). To further explore the characteristics of TLR3 in yellow catfish, the conservation of the amino acid sequences in different species of teleosts was analyzed, and our results showed that there was a highly conserved TIR domain in the C-terminal of these TLR3 proteins (Figure 1). Our results revealed a high conservation among TLR3s of different fishes. Furthermore, we observed that two α-helixes, nine η-helixes and 26 β-sheets were shared among these TLR3 proteins (Figure 1).

### 3.2. Gene Structure, Collinearity and Protein Structure Analysis

To compare the gene structures among different TLR3 genes, a comparative analysis was conducted. There were five exons—exon 1 (584 bp), exon 2 (1855 bp), exon 3 (194 bp), exon 4 (569 bp) and exon 5 (346 bp)—in the gene structure of TLR3 in yellow catfish. Introns 1, 3 and 4 are 234, 910 and 121 bp, respectively, while intron 2 is a comparatively large intron with the size of 1196 bp. There were five exons and four introns in TLR3 genes from Teleostei, Reptilia and mammals (Figure 2), indicating that these gene structures were highly conserved in the three taxa. However, the length of the third exon in teleost TLR3s was 194 bp, which is slightly different from the 191 bp in Reptilia and mammals.

Our results showed that TLR3 (XM_027134964) is located at 16210396–16217622 of chromosome 7 (NC_062524.1) in the yellow catfish (GCA_022655615.1) genome. In addition, a comparative analysis was conducted to uncover the genetic diversity of TLR3 in various teleosts. As shown in Figure 3, a conserved gene cluster, *FATacla*–*mr1Aa*–*P450*–TLR3–*sorbin*–*PDZ*–*npy2r*, existed in nearly all the genomes of representative species. It may suggest a highly conserved synteny of TLR3 genes across vertebrates.

Our prediction about the protein spatial structures of TLR3s showed that yellow catfish TLR3 is highly similar to that in the red flour beetle (*Tribolium castaneum*), sea lamprey (*Petromyzon marinus*), thorny skate (*Amblyraja radiata*), toad (*Bufo bufo*), three-toed box turtle (*Terrapene carolina triunguis*) and mouse (*Mus musculus*). TLR3s in these six species are horseshoe-shaped and the TIR domains are on the top of a nonclosed torus (Figure 4). However, TLR3 in the common pheasant (*Phasianus colchicus*), a representative bird, is a closed ring with multiple β-sheets (Figure 4).

### 3.3. Phylogenetic and Selection Analyses

The phylogenetic analysis of TLR3s in 64 selected species demonstrated that these homologous genes can be divided into two groups, invertebrates and vertebrates. The results also revealed that yellow catfish TLR3 was in the same subgroup as other teleosts. Among the bony fishes, the yellow catfish (*Pelteobagrus fulvidraco*), the goonch catfish (*Bagarius yarrelli*) and channel catfish (*Ictalurus punctatus*), all of which belong to Siluriformes, were clustered in this study. Obviously, the vertebrate TLR3 group was further clustered into six subgroups including mammals, amphibians, birds, reptiles, cartilaginous fishes and bony fishes, while the invertebrate group only contains arthropods (Figure 5). It is found that the TLR3s in bony fishes were close to those in arthropods, and they appeared after the evolution of cartilaginous fishes.

To analyze the structure of TLR3s in different fishes, the SMART software was used. The functional domain prediction showed that TLRs contain seven domains individually, including a low-complexity region, leucine-rich repeat (LRR), leucine-rich repeat C-terminal (LRR_CT), leucine-rich repeat N-terminal (LRR_NT), leucine-rich repeat (LRR_TYP) typical (most populated) subfamily, TIR domain and transmembrane region (TR). Previous studies have shown that the concave structure of the LRR domain plays an important role in the recognizing of different PAMPs, and the LRR_CT at the C-terminus of the LRR domain stabilizes the protein structure [17]. Moreover, TIR domains are indispensable in TLR-involved signaling [17]. Among these domains, however, TIR is absent in most TLR3 proteins of Arthropoda, and TR is also deleted in some arthropod TLR3s.

Among the values of dN/dS from seven examined taxa, those of TLR3s in Arthropoda were the highest (Figure 6). Except for Osteichthyes, the average dN/dS of TLR3 was higher than that of corresponding TLR3-TIR, implying that TLR3 might have evolved earlier than TLR3-TIR in the six taxa (Figure 6). TIR domains of TLR3 in Reptilia, Chondrichthyes, Amphibia, Mammalia and Arthropoda exhibited the lowest rate of evolution (with the average dN/dS values less than 0.1 or close to 0.1), implying that TLR3-TIR is highly conserved in these classes during the process of evolution. The evolutionary rates of the TLR3-TIR domains are significantly different from TLR3. Interestingly, amphibian TLR3s had the lowest averages of dN/dS, although the dN/dS values of TLR3s and TLR3-TIRs in all taxa were less than 1.

### 3.4. Distribution of TLR3 in Different Tissues in Hybrid Yellow Catfish

Quantitative real-time PCR was used to analyze the distribution of TLR3 in different tissues in hybrid yellow catfish. As there are no available reported studies about the tissue expression of TLR3 in hybrid yellow catfish, the mRNA transcription levels of TLR3 were analyzed here. Our results showed that TLR3 was extensively transcribed in the 12 sampled tissues (including caudal fin, barbel, gallbladder, intestine, liver, stomach, heart, gills, spleen, kidney, skin and brain), but exhibited a tissue preference (Figure 7). The transcription level of TLR3 was highest in the liver and intestine, while lower expression levels of TLR3 were found in the barbel, gallbladder, stomach, heart, gills, spleen, kidney, skin and brain, and the lowest level of TLR3 was in the caudal fin (Figure 7). These results indicated that expression of TLR3 in hybrid yellow catfish is not only restricted in the tissues involved in the immune response, but also in nonimmune tissues, raising the possibility that TLR3 exhibits multiple functions in addition to the immune regulating roles.

### 3.5. Effects of Bacteria and Poly (I:C) on the Transcription Levels of Hybrid Yellow Catfish TLR3

To investigate the potential roles of hybrid yellow catfish TLR3 in the response to exogenous immune stimuli, we measured the transcriptional levels of the hybrid yellow catfish TLR3 gene among multiple tissues (including gill, liver, spleen and kidney) after the *A. hydrophila* or poly (I:C) challenge.

For *A. hydrophila* challenge, the transcription of hybrid yellow catfish TLR3 was increased to a significantly higher degree in the spleen after the bacterial infection for 24 and 48h (Figure 8a). In the kidney, the expression of TLR3 was significantly induced at 3, 6, 12 h and the highest expression levels were at 12 h (Figure 8b). Then, the expression levels decreased to the control level gradually (Figure 8b). For the poly (I:C) challenge, however, the transcription of hybrid yellow catfish TLR3 declined significantly at 3 h in spleen, and then exhibited relatively low levels at 6 and 12 h; subsequently, the transcription level increased with the challenge time, and the peak value was reached at 24 h and then dramatically declined at 48 h (Figure 8c). Similarly, the transcription of TLR3 declined after poly (I:C) stimulation in the liver at 3 h, then declined to the lowest value at 6 h, increased significantly again at 12 h and peaked at 24 h, then subsequently declined at 48 h (Figure 8d).

### 3.6. Analysis of the Binding Site of Poly (I:C) and dsRNA Virus between TLR3-ECD

Docking of the 3D model including a poly (I:C) segment and yellow catfish TLR3-ECD using HADDOCK showed that R103, Q79, W150, E173, R200, Q224, R251, K326, K328, Y357, N383, E361, N412, W386, H439, S463, Q466, N515, K536, H529, N537 and Q538 were identified as the potential sites of H-bond formation with poly (I:C) (Figure 9a). Interaction of poly (I:C) with loach TLR3-ECD showed that G238, R240, Q291, D265, Q366, W399, Q370, R371, S503, S504, R529, N555, S554, Q507, H579, K576, Y611 were identified as the potential binding sites (Figure 9b). Furthermore, docking of the 3D model including a poly (I:C) segment and the mouse TLR3-ECD showed that K223, S250, N276, Q252, Q279, R302, K331, Y384, S388, N414, D438, Y466, R485, R489, Q539 were identified as the interacting sites (Figure 9c). The binding scores of these complexes above obtained through the Zdock software (https://zdock.umassmed.edu/, accessed on 11 August 2022) were 1867.67, 1640.30 and 1722.36, respectively. Previous research showed that scores above 1000 usually indicate the binding is tight between two molecules, which further proved that TLR3 in yellow catfish, loach and mouse binds firmly to the poly (I:C).

Prediction of the 3D model between the dsRNA virus and yellow catfish TLR3-ECD showed that R134, R136, R155, K157, D181, K210, K232, D257, Q259, K285, N287, N309, K311, T334, K335, H337, T338, S339, A340, L341, R362, S388, T389, K392, G417, K419 and Q466 were identified as the potential binding sites (Figure 10a). Docking of the 3D model between the dsRNA virus segment and the mouse TLR3-ECD showed that K181, K183, L185, R188, S207, N208, P209, K211, N231, A232, Q233, Q260, Y284, N286, Y308, N310, R332, K389, K417, Y466, N467, K468, R490, A492, K494, N516, N518, A520, N521, H540 and R545 were identified as the interacting sites (Figure 10b). These results indicated that teleost TLR3s may have more active sites than mammalian TLR3s, and teleost TLR3 proteins may recognize more types of viruses.

## 4. Discussion

In this study, TLR3 in yellow catfish was identified and characterized using comparative genomic approaches. Sequence alignment and secondary structure prediction showed that TLR3s in teleosts shared two α-helixes, nine conserved η-helixes and 26 β-sheets, and the sequence similarity of TR and TIR domains among teleost TLR3s is over 60% (Figure 1). Gene structure analysis showed that TLR3s individually contained five exons and four introns in teleosts, reptiles and mammals, and the third exons in teleosts are the same length of 174 bp (Figure 2). Genome collinearity analysis revealed that seven genes (including *FATacla*, *mr1Aa*, *P450*, TLR3, *sorbin*, *PDZ* and *npy2r*) consisted of a specific cluster with a conserved arrangement in various teleost genomes (Figure 3). Protein spatial structure prediction showed that TLR3s were usually horseshoe-shaped in representative species, except for birds (Figure 4). Phylogenetic analysis showed that the TLR3 topology is divided into two subgroups of vertebrates and invertebrates (Figure 5), and the vertebrate TLR3s were further clustered into six classes of mammals, amphibians, birds, reptiles, cartilaginous fishes and bony fishes (Figure 5). The natural selection analysis exhibited that the dN/dS values of TLR3s and TLR3-TIR domains of all taxa were less than 1 (Figure 6). Quantitative RT-PCRs showed that the transcript of TLR3 was widely distributed in all tested tissues in hybrid yellow catfish, among them the highest occurred in the liver, and the lowest appeared in the caudal fin (Figure 7). Additionally, the transcription levels of TLR3 in spleen and kidney were upregulated significantly after exogenous *A. hydrophila* infection (Figure 8a,b). Furthermore, the transcription of TLR3 in spleen and kidney showed a change pattern with a wave shape after the poly (I:C) infection (Figure 8c,d).

Prediction of the secondary structures showed that TLR3s shared two α-helixes, nine conserved η-helixes and 26 β-sheets in various teleosts (Figure 1). In addition, TR and TIR domains among fish TLR3s were highly similar (more than 60% similarity; Figure 1), suggesting modulation of similar physiological processes by these proteins. Comparative gene structure analysis showed that the TLR3 genes consisted of five exons and four introns in teleosts, reptiles and mammals (Figure 2), and their third exons (191–194 bp) were similar in length, demonstrating that TLR3 is very conservative during evolution.

A cluster composed of seven genes (*FATacla*–*mr1Aa* –*P450*–TLR3–*sorbin*–*PDZ*–*npy2r*) was identified for the first time (Figure 3a), although the divergence of bony vertebrates experienced chromosomal rearrangements approximately 465 million years ago [44]. This cluster in teleosts is always a single copy, reconfirming that TLR3 is conserved with functions in similar biological processes among various teleost species. Interestingly, an unexpected inversion of the common *npy2r*–*PDZ*–*sorbin*–TLR3–*P450*–*mr1Aa*–*FATacla* cluster was observed in a few fishes, such as spiny chromis damselfish (*Acanthchromis polyacanthus*), Siamese fighting fish (*Betta splendens*) and slender sharksucker (*Echeneis naucrates*; Figure 3a,b). Conserved collinearity was found in the TLR1 and TLR19 of brown trout, rainbow trout and zebrafish [45], suggesting that upstream as well as downstream syntenic gene blocks of *TLRs* were conserved in fish.

The crystal structure of the ectodomain in mammalian TLR3s was first resolved in humans, and the mammalian TLR3s were commonly predicted with a large horseshoe-shaped solenoid with 23 LRRs [46]. dsRNAs bind to TLR3 dimers, which contain three interaction sites in each ectodomain, and one of the two binding sites was responsible for the dimerization [47]. Our structural predictions supported the high conservation of TLRs among diverse mammals and teleosts (Figure 4), implying that TLR3s may play similar biological functions in these animals. However, the TLR3 in the common pheasant, a representative of birds, is a ring with multiple β-sheets that is remarkably different from the common horseshoe-shaped structure in other species (Figure 4), suggesting functional divergence of avian TLR3s during species evolution.

As the constructed phylogenetic tree of TLR3s contained two distinct subgroups of invertebrates and vertebrates (Figure 5), vertebrate TLR3s could be divided into six classes. Invertebrates contain only Arthropoda, similar to those studies reported previously [48,49,50], implying that TLR3 proteins in various fishes might play a role in similar biological processes although they evolved independently throughout their natural history. Based on the results of the TLR3 phylogenetic tree constructed using two methods (maximum-likelihood method and neighbor-joining method), we speculated that TLR3 may have a unique evolutionary process, indicating that the evolution rate of TLR3 proteins in different taxa is different. TLR3 in cartilaginous fishes maintains similar functions to TLR3 in tetrapods and bony fishes. There are some contrasting phenomena in the evolution and function of genes in cartilaginous fishes. The specific evolution process needs to be further studied. Analysis of natural selection showed that all the dN/dS values of TLR3s and TLR3-TIR domains in the examined taxa were less than 1 (Figure 6), indicating that purifying selection had dominated the evolutionary patterns of TLR3s and TLR3-TIRs [51]. Since the phylogenetic tree is mostly consistent with the species tree, with average dN/dS values far less than 1, the evolution of TLR3 was conservative in vertebrates. In addition, the slightly greater average dN/dS values of TLR3 and TLR3-TIRs in vertebrate classes (including Mammalia, Amphibia, Aves, Chondrichthyes and Reptilia) and an invertebrate class (Arthropoda) (Figure 6) suggested that the evolving rates among these classes were uneven. Interestingly, Osteichthyes (teleost fish) have a higher average dN/dS value for TLR3-TIR than TLR3 (Figure 6), implying that TLR3-TIR might have evolved more quickly than TLR3 in these species, which may help to maintain the stability of corresponding biological functions. The dN/dS value of TLR3-TIR in Arthropoda (arthropods) and Chondrichthyes (cartilaginous fish) is the lowest among all the analyzed species, suggesting that the function of TLR3 protein is very conserved in lower classes (Figure 6). These findings therefore improve our understanding of TLR3 in vertebrates, especially for that in teleosts. It is interesting to uncover the evolutionary changes among conserved domains of TLR3. Although no significant differences among TLR-TIR homologous domains from the majority of classes based on different methods were found in this study, an obvious difference in average dN/dS values was identified between all TLR3s and TLR3-TIRs (Figure 6). This indicates that the natural selection between all TLR3s and TLR3-TIRs was totally different in hybrid yellow catfish.

Quantitative real-time PCRs were performed to investigate the tissue distribution of hybrid yellow catfish TLR3. Similar to the TLR3 in channel catfish [52], the transcription of TLR3 was widely present in all sampled tissues (Figure 7) although the highest value existed in the liver. This pattern is consistent with the report on golden pompano [19], implying that TLR3 might participate in immune responses in different fishes. Previous studies showed that TLR3 mRNA in large yellow croaker (*Pseudosciaena crocea*) was extensively transcribed and the three highest transcriptions were determined in liver, intestine and heart [53]. This tissue pattern is similar to that of the TLR3 in the present research (Figure 7). Similarly, TLR3 exhibited the highest expression in the liver of rainbow trout (*Oncorhynchus mykiss*; [54]). We therefore propose that fish liver might play an indispensable role in the synthesis of TLR3.

*A. hydrophila* isolated from certain local freshwater ponds has been widely used in previous infection studies [55]. Various fishes, such as channel catfish, bass and yellow catfish in freshwater, are sensitive to the infection of this bacterium [56,57,58]. Interestingly, the transcription of hybrid yellow catfish TLR3 changed significantly after the infection of exogenous *A. hydrophila*, and the most sensitive tissues included spleen and kidney (Figure 8a,b). However, no such upregulation of TLR3 transcripts was found in zebrafish when infected by the Gram-positive *Mycobacterium marinum* [59], while the expression of TLR3 increased after the infection of Gram-negative *Edwardsiella tarda* in zebrafish [60]. Mixed responses were also found in zebrafish after an exposure to lipopolysaccharide [61]. Such stimulation was also observed in mammals. For example, the cell wall component of Gram-negative bacterial lipopolysaccharide triggered the expression of murine TLR3 [62]. These results suggest that TLR3s in teleosts may play a similar role to the mammalian TLR3s in response to exogenous bacteria.

It has been reported that metabolic wastes, toxins and drugs from fish bodies are removed by the kidney, and circulating cytokines as well as lipopolysaccharide are recovered by this organ for collection of blood-derived proteins to maintain the immune system in fishes [63]. Previous research found that the spleen is an lymphoid organ with indispensable functions in the immune response and hematopoiesis as well as clearance of red blood cells [64]. In these biological processes, the spleen combines the innate and adaptive immune systems in a unique organization to effectively remove microbial and cellular debris from the blood [65]. It has been well recognized that TLR3 recognizes dsRNA poly (I:C) to activate the innate immune system through generating proinflammatory cytokines and chemokines with key functions [66]. To examine the functions of TLR3 in teleosts, we determined the transcriptional changes in hybrid yellow catfish TLR3 after poly (I:C) stimulation in spleen and kidney. It first declined and then increased to a peak at 24 h followed by declining after poly (I:C) challenge for 48 h (Figure 8b,c), indicating that TLR3 may regulate the immune response triggered by dsRNA virus infections in spleen and kidney. Likewise, the highest expression level of TLR3 was found at 24 h after poly (I:C) challenge in mammals [54]. In addition, an upregulation of chicken TLR3 in spleen at 24 h after a virus infection was determined in a previous report [67], which is consistent with our present results in yellow catfish. These data imply that TLR3 may have similar functions in mammals, birds and teleosts in response to exogenous dsRNA viruses. As observed in other fishes, the expression of TLR3 in the kidney of Prussian carp (*Carassius auratus gibelio*) reached a peak at 12 h after infection with cyprinid herpesvirus 2 (a dsDNA virus; [68]), and the transcript of TLR3 in golden pompano (*Trachinotus ovatus*) kidney also increased and first peaked at 6 h in response to a poly (I:C) stimulation [19]. However, the expression patterns of TLR3 in different fishes are not the same after infection with viruses. This may result from slight differences in developmental status, virus type and/or virus dosage.

Poly (I:C) is a TLR3 ligand and an analog of viral dsRNA. Previous studies have shown that TLR3 recognizes double-stranded RNA virus in mammals to trigger downstream immune responses [41,47]. However, few similar studies have been carried out in teleost fish. In our current research, TLR3 in yellow catfish and loach was used for molecular docking with poly (I:C) and a double-stranded RNA virus. The horseshoe-shaped solenoid structure of TLR3-ECD in yellow catfish resembles that in the mouse, indicating a potentially similar type of ligand recognition and interaction in these two species (Figure 9). Our results also showed that TLR3 in teleosts forms a complex with poly (I:C) (Figure 9a,b), which is consistent with those in higher mammals (Figure 9c), suggesting that poly (I:C) may play the same role in the immune system in teleosts as in higher mammals. The docking results of dsRNA and TLR3 are consistent with that of poly (I:C) and TLR3 (Figure 9 and Figure 10). Simultaneously, the relatively large number of binding site functions in the interaction between dsRNA and TLR3 in yellow catfish compared to that in the mouse may indicate the possibility of a broader dsRNA recognition region in teleost TLR3.

## 5. Conclusions

This is a comprehensive investigation and systematic comparison of TLR3 genes in various teleost species. The evolution of TLR3 seems to be highly conserved, and transcriptional analysis showed that the TLR3 gene was expressed in the 12 examined tissues. The expression analyses showed that the TLR3 gene was significantly enhanced in spleen and kidney after the *A. hydrophila* or poly (I:C) stimulation, indicating that TLR3 may participate in the immune processes against both bacterial and viral pathogens in teleosts. Molecular docking results showed that dsRNA virus is a potential ligand of TLR3 proteins in teleosts. In spite of being such a preliminary work, our present study provides good genetic support for better understanding of the immune mechanisms of teleost fish when infected with exogenous bacterial and/or viral pathogens.

## Figures and Tables

**Figure 1 animals-13-00288-f001:**
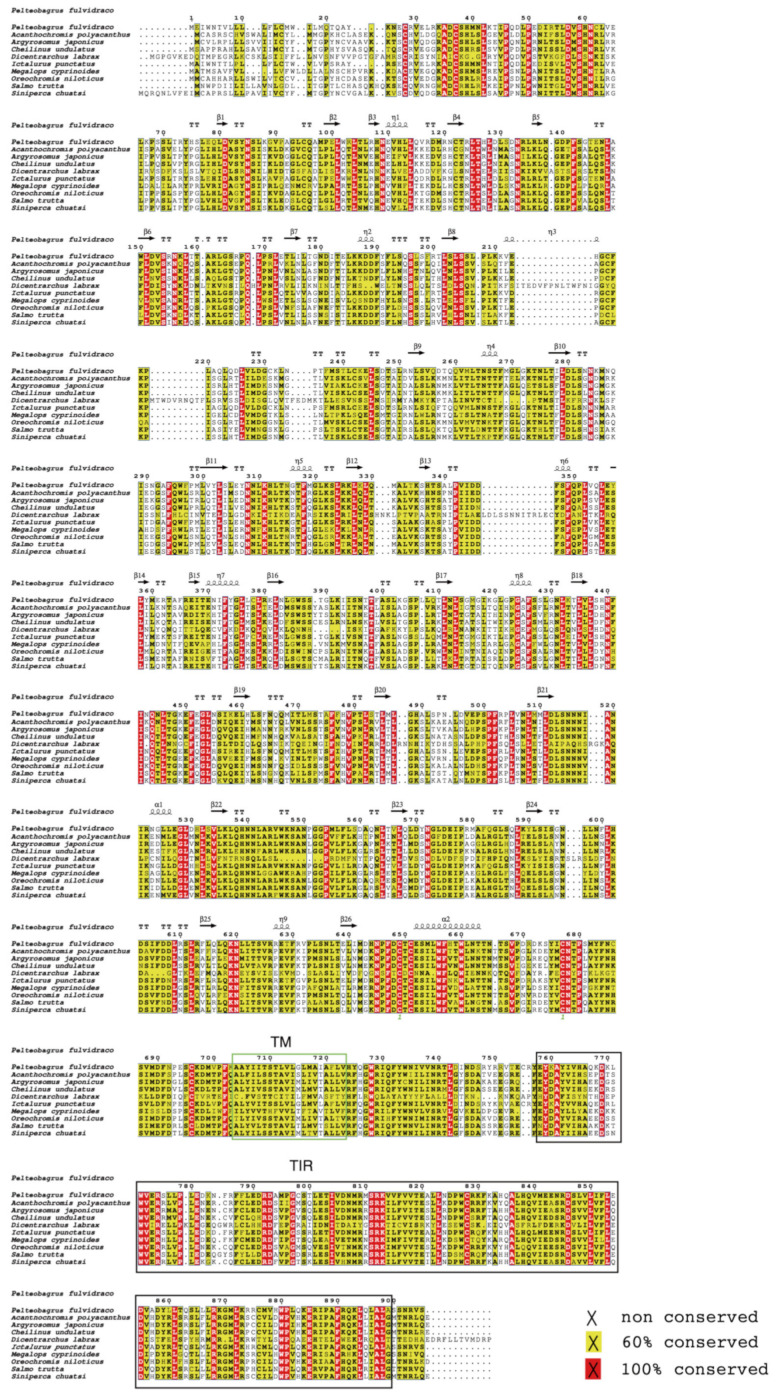
Multiple sequence alignment of teleost TLR3s. The yellow areas represent high similarity (more than 60%), and the red areas represent the same residues. Sequences within green boxes are transmembrane (TM) regions, and those in black boxes are Toll/interleukin-1 receptor (TIR) domains.

**Figure 2 animals-13-00288-f002:**
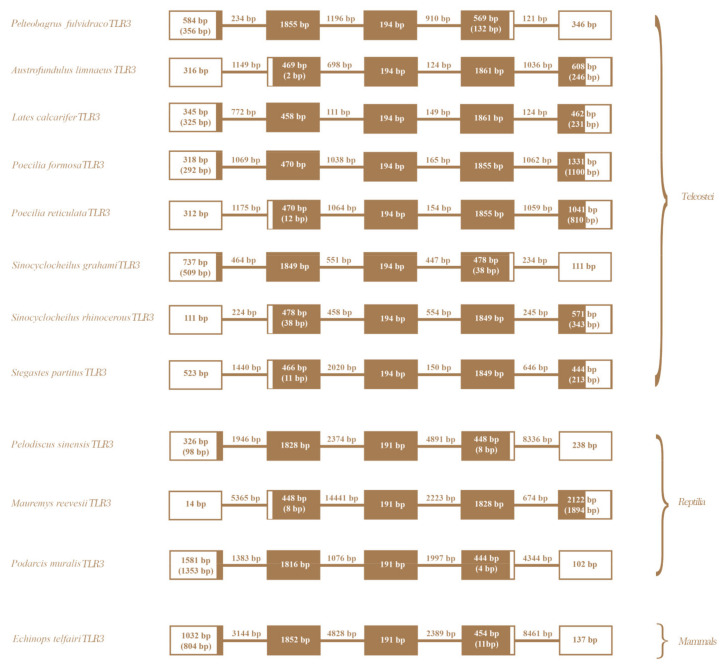
Gene structures of TLR3s in various vertebrate species. Boxes and lines represent exons and introns, respectively. Residue numbers are listed inside the boxes, indicating the high similarity of TLR3s among different vertebrates.

**Figure 3 animals-13-00288-f003:**
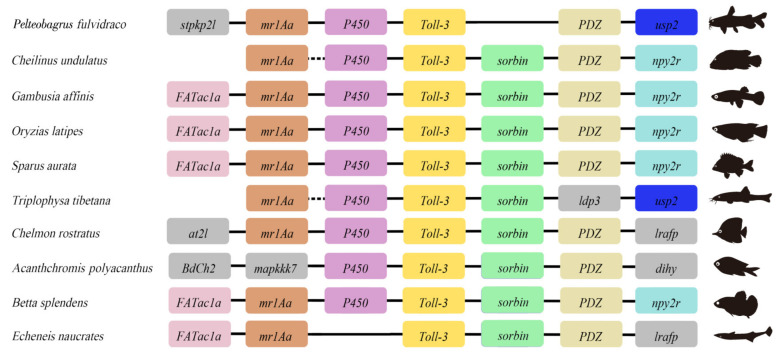
Comparative genomic synteny of TLR3 among representative teleosts. The colored boxes represent genes, and solid lines stand for intergenic regions. Genes with conserved synteny are marked with the same colors.

**Figure 4 animals-13-00288-f004:**
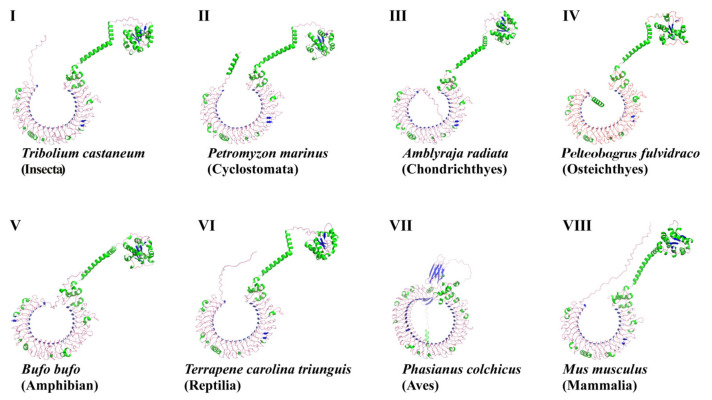
Spatial structures of TLR3 proteins in eight representative species of eight examined taxa. The green parts are helixes, the blue parts are sheets and the pink parts are random curls. GenBank accession numbers of these protein sequences are listed in Appendix A.

**Figure 5 animals-13-00288-f005:**
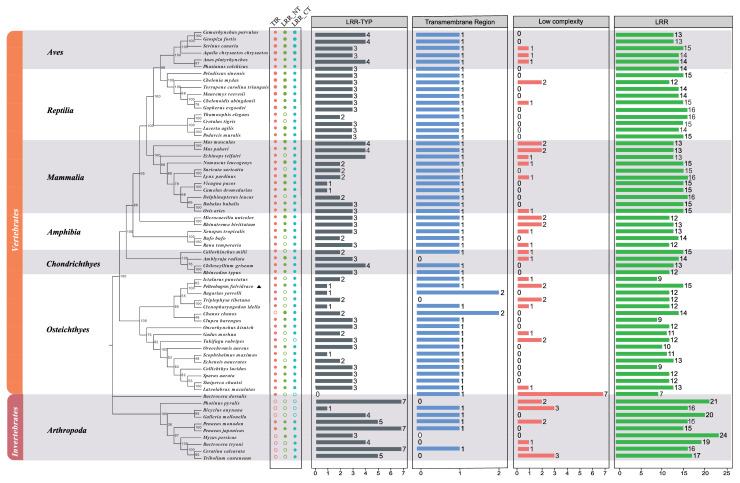
A comprehensive phylogenetic tree of TLR3. This tree was constructed with 59 selected protein sequences using the MEGA X program. The yellow catfish is marked with an asterisk. Oriental fruit fly (*Bactrocera dorsalis*) is selected as the outgroup. Protein sequence IDs are provided in Appendix A.

**Figure 6 animals-13-00288-f006:**
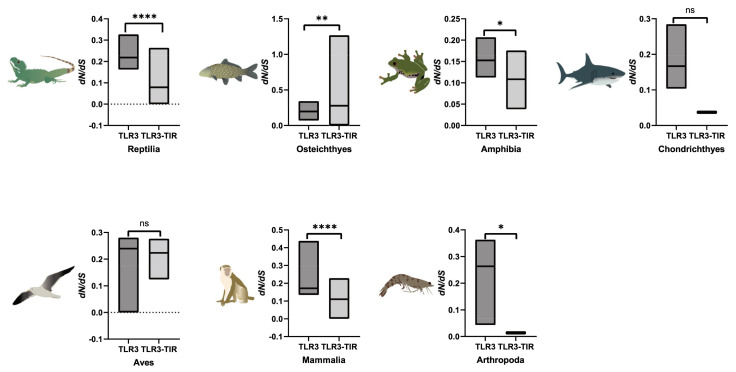
Natural selection analyses of TLR3s and TLR3-TIRs in seven animal taxa. Asterisks indicate significant differences (*p* < 0.05). When *p* < 0.05, significant, marked with “*”; when *p* < 0.01, extremely significant, marked with “**”; when *p* < 0.0001, the degree of significance is extremely high, marked with “****”.

**Figure 7 animals-13-00288-f007:**
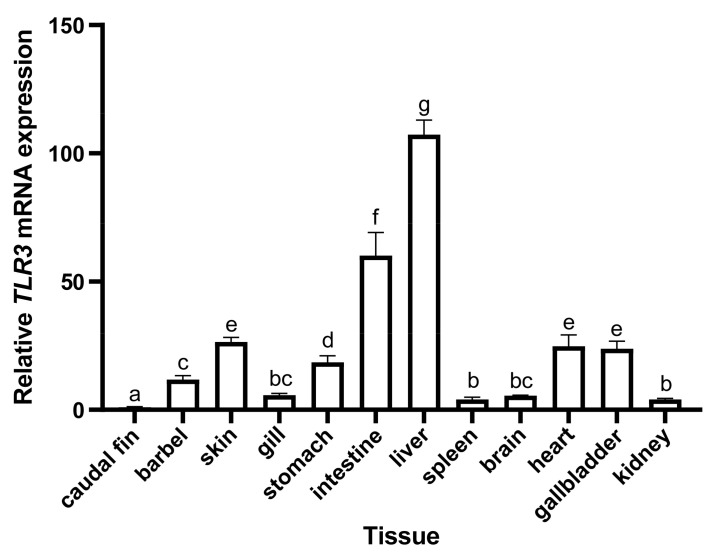
Tissue distribution pattern of TLR3 in hybrid yellow catfish. Twelve tissues were collected in the present study. Data are expressed as relative expression levels with normalization by the *β-actin* gene. Error bars represent standard error of the mean (SEM; *n* = 3). Defferent letters upon the bars represent significant difference among groups.

**Figure 8 animals-13-00288-f008:**
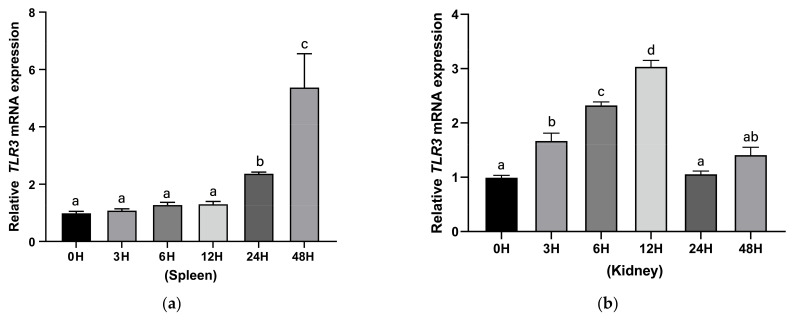
Relative expression levels of TLR3 in different tissues after *A. hydrophila* or poly (I:C) challenge. (**a**,**b**) Transcriptional changes in hybrid yellow catfish TLR3 in spleen and kidney at 0, 3, 6, 12, 24 and 48 h after *A. hydrophila* challenge. (**c**,**d**) Inductive transcription of hybrid yellow catfish TLR3 in spleen and kidney at 0, 3, 6, 12, 24 and 48 h after the infection with poly (I:C). Data are represented as mean ± SEM (*n* = 3). Groups with significant differences are marked by different letters above error bars.

**Figure 9 animals-13-00288-f009:**
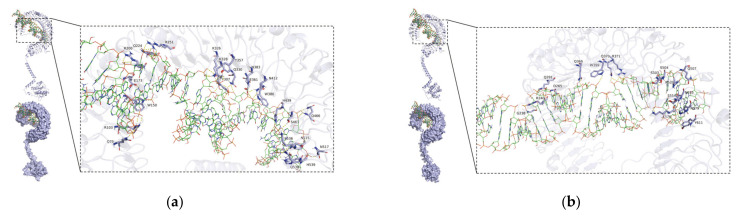
A 3D model of interaction between poly (I:C) and TLR3-ECD predicted/constructed using HADDOCK program. TLR3-ECDs are shown in a “U”-shaped structure. The right side of the figure refers to the magnification of the three docking sites, respectively. (**a**) Docking of TLR3 in yellow catfish and poly (I:C). (**b**) Docking of TLR3 in loach and poly (I:C). (**c**) Docking of TLR3-ECD TLR3 in mouse and poly (I:C).

**Figure 10 animals-13-00288-f010:**
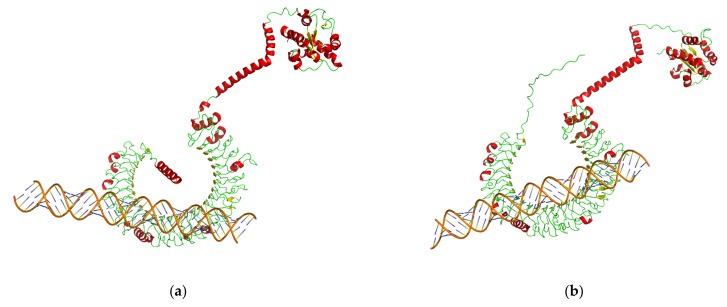
A 3D model of interaction between dsRNA virus and TLR3-ECD predicted/constructed using HADDOCK program. (**a**) Docking of TLR3 in yellow catfish and dsRNA virus. (**b**) Docking of TLR3 in mouse and dsRNA virus.

**Table 1 animals-13-00288-t001:** Primers used for the quantitative RT-PCRs.

Primer Name	Primer Sequence (5′-3′)	Amplicon (bp)
beta-actin F	GGACCAATCAGACGAAGCGA	105
beta-actin R	TCAGAGTGGCAGCTTAACCG	
Toll-3 F	CCTGTTGCAAGTCCGAGACA	119
Toll-3 R	CCAGCCAGGCCAGATTCTCT	

**Table 2 animals-13-00288-t002:** Amino acid similarity of yellow catfish TLR3 to other teleost TLR3s. To determine the percentage similarity, the TLR3 sequence of yellow catfish was aligned with other teleost orthologues using CLUSTAL W multiple alignment.

Species	Protein Accession Number	Percentage Similarity (%)
*Acanthochromis polyacanthus*	XP_022067302.1	65.5%
*Argyrosomus japonicus*	QOS44501.1	66.8%
*Cheilinus undulatus*	XP_041642339.1	65.8%
*Dicentrarchus labrax*	CBN82176.1	38.2%
*Ictalurus punctatus*	ABD93873.1	90.1%
*Megalops cyprinoides*	XP_036372873.1	66.1%
*Oreochromis niloticus*	XP_025764171.1	65.8%
*Salmo trutta*	XP_029605762.1	69.8%
*Siniperca chuatsi*	QEU52192.1	66.1%

## Data Availability

The original contributions presented in the study are included in the article/Appendix A. Further inquiries can be directed to the corresponding authors.

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
