# Peer review of "Toll-like Receptor 3 in the Hybrid Yellow Catfish (*Pelteobagrus fulvidraco ♀ × P. vachelli ♂*): Protein Structure, Evolution and Immune Response to Exogenous *Aeromonas hydrophila* and Poly (I:C) Stimuli"

_animals, 2023, doi:10.3390/ani13020288_

Round 1

Reviewer 1 Report

The manuscript submitted by Guo et al. explored the protein structure and evolution pattern of TLR3 in teleosts, and also investigated the expression pattern of TLR3 after the Aeromonas hydrophila and poly(i:c) stimulation at different time point, and finally demonstrated the role of TLR3 in the recognition of viral pathogens through the molecular docking. This study provides new insights into the molecular mechanism of viral pathogens infections in teleosts. The structure of the manuscript is well organized, the language logic is clear, the pictures and tables are self-explanatory, and the whole level of this manuscript meets the reception requirements of this journal. Thus, I recommend this manuscript can be considered for publication after minor revision.

1.       Change the order of Table 2 and Table 3 to maintain the logic of this manuscript.

2.       Please add more descriptions about the expression pattern and molecular mechanism such as tissue distribution and molecular docking in the conclusion section.

3.       What are the strengthens and weaknesses of the present study?

Author Response

Thank you very much for your high praise of our manuscript, we have promoted our manuscript according to your comments. Please refer to the PDF file for details.

Reviewer 2 Report

The paper describes the isolation and functional characterization of the tlr3 in innate immune response after challenge with bacteria and a virus-like molecule in the Siluriform yellow catfish which is an aquaculture important species in China. The tlr3 genes is a single gene receptor in teleosts with orthologues in tetrapods known to be responsive to virus or viral-like infections. The study is interesting but not scientifically sound in my point of view to the journal as is just another descriptive study of a tlr3 gene from another teleost fish which does not bring anything novel in relation to what is already known. There are many confusions in the text with data interpretation, and some of the analyses does not seem to be corrected and for some not sufficient detail is provided. Therefore, i am forced to reject the manuscript. Nonetheless I provide a list of several comments that in my view can help the authors to improve the study.

Line 70 “As we know, teleosts represent a transition point in the phylogenetic spectrum between invertebrates and mammals” this sentence is not correct as teleosts. What about the other fishes? Teleosts are diverse group of ray-finned fishes.

Line 83- “Simultaneously, findings showed that TLR3 in Japanese sea perch functions in specific regions” – please clarify and be more specific.

Line 86- What do you mean by “danger signals”? Infection response to pathogen invasion? Be more specific please.

Line 105- Why citing the authors own work in the introduction after describing the strategy taken to support that the methodology here employed is similar to a previous study? Please remove.

Line 119 – Please provide the SRA number of the submitted transcriptome data used in the analysis

Line 127- Please remove reference to your previous work in the methods because they are standard procedures used for comparative genomics studies. Please include the length of the region used for analysis (how many neighbor genes upstream and downstream of tlr3) have you considered in your analysis and how did you obtained/confirm their identities. Which species genome have you used as reference to search for the homologous genome regions?

Line 131- What were the comparative genomics is a series of methods. Please specify which ones you have used in your analysis.

Line 140- “.. as described previously” is not correct. You have used the SMART tool to identify the protein motifs.

Line 144- What do you mean by “determine the origin of the tlr3 gene in vertebrates”? It is already known that tlr3 genes are conserved across many vertebrate clades from Agnathan to Mammalian. So its emergence occurred very early in the vertebrate evolution. 

Line 148- “ evolutionary landscape” please clarify?

Line 151- “infection experiments” or “immune challenges?” Please also verufy in the rest of the text.

Line 158- Not “accommodation” it should be “acclimatization”

Line 162- Poly(I:C) does not provoke a “virus infection” because is not a virus but a virus-like molecule. Moreover, this will not be an infection but a challenge.

This section is confusing because at the beginning the authors say that they have collected 12 tissues from 3 animals after the immune challenges but then subsequently in the text they only collected immune related tissues for the tissue expression analysis. Please revise.

Line 175- It is important to know is the PCR products were sequenced to confirm their identity. Also give a bit more detail on RNA extraction and cDNA synthesis were performed. How much tissue was used for the RNA extraction, how much tRNA was used for the cDNA synthesis? Was the RNA treated with DNAse to remove for potential genomic contamination? Please provide more detail on the methodologies applied. What was the qPCR cycle used? Where negative controls performed? Please also do not refer to a previous publication but describe the methods you have used to quantify the changes in the transcript expression.

Line 188- is this 46 bp molecule a virus?

Line 204- I am not totally sure if all these sequences correspond to isolated mRNA transcripts. They must be predicted coding sequences available from the databases and not cDNAs. How were the percentage similarities calculated that are represented in Table 2 (which is missing and mislabeled as Table 3)? Please include this information in the methods. Legend of table 3: “Amino acid sequence similarity…”

Line 205- Please clarify this sentence because is mixing nucleotides and orf in amino acids. Figure 1 does not clearly indicate the identity/similarity between the sequences of the different fishes it just represents the sequence alignment. It is also not surprising that the yellow catfish is more similar to the channel catfish because they belong to the same order. Why also describing the % of similarities in the text if there is a table where they are described? 

Line 220 “These levels of sequence homology might be explained by the fact that TLR3 sequences in other ray-finned fishes are underrepresented.” Please delete this sentence as it makes not much sense. Like other TLRs the TM and the TIR domains are also the most conserved receptor regions in comparison with the N-terminal that contain the Leucine-rich repeat (LRR) domains.

Figure 1: The red areas represent the conserved residues. How were the coloured code regions achieved?

Table 2: What is the reason for this table. It does not add any significance to the paper. It just described general sequence characteristics and not directly related with the characterisation of the teleost tlr3. If the authors would like to maintain can be moved to supplementary information.

Line 236- why detailed description of size of introns and exons is necessary? What is the relevance of Figure 2 for the main manuscript. This figure can be moved to supplementary.

Figure 3. It is not clear if we are looking at short or long synteny as not description of the gene positions or of the length of the represented genome fragments is indicated. Please modify the figure. Please indicate what is a) and b)? Perhaps to be more informative and as to study the evolution of the tlr3 in vertebrates you should include other representatives of other vertebrate clades and remove some of the teleosts as they are all the same and not evolutionary informative to understand how tlr3 gene evolved in vertebrates. What does the arrow and 180º mean? The data represents the results of the gene synteny analysis not of a genome collinear analysis as indicated in many parts of the text.

Figure 5-  The tree lacks the bootstrap values. Is there any evidence that the arthropod tlrs represented are the ancestral-like genes and the orthologues of the vertebrate tlr3? The clustering of the sequences in the tree does not seem correct as the cartilaginous fish tend to cluster closer to the Amphibian and should radiate prior to the teleost. Tlr3 has also been isolated from lampreys and is not included in the analysis.

Line 271- This is not surprising….” The results also revealed that yellow catfish TLR3 was in the same subgroup as other teleosts.” You should be more specific and describe with which species it cluster more closely. 

The interpretation of the tree is totally incorrect and goes against what is known about the evolution of the species. “it is found that the TLR3s in bony fishes were closed to those in arthropod, and they appeared after the evolution of cartilaginous fishes.”

Author Response

Thank you for your detailed comments on our study, your questions and suggestions have greatly improved our manuscript. We have made a one-on-one reply to your question and revised our manuscript accordingly, please refer to the PDF file for the specific reply.

Reviewer 3 Report

My comments are here

Authors ignored an important work done on TLR-3 https://www.mdpi.com/1422-0067/21/11/3755

Please discuss your findings with it.

Line 159, how were fish confirmed as specific pathogen free/healthy before sampling?

Line 162, which virus infection do you mean?

Line 165, why were fish injected intraperitoneal with A. hydrophila instead of bath exposure. In nature, the immune system will be different rather than           laboratory conditions. Please explain it.

Line 176, did they perform DNase treatment during RNA extraction? RNA extraction protocol is not explained. What RNA concentration used to prepare cDNA synthesis?

Line 180, details of quantitative PCRs protocol and reaction are missing. How were qPCR data analysed?

Table 1: add PCR product size of each primer set.

Result section, lines 202-227: Was TLR3 single copy? What about the location of TLR3 on chromosome?

Author Response

Thank you very much for your approval and constructive feedback of our manuscript. We have responded to your questions one by one and revised our manuscript accordingly. Please refer to the PDF file for details.

Round 2

Reviewer 2 Report

The manuscript has been substantially improved and the authors have addressed all my suggestions. I only have two minor comments:

a) line 172- “…and the virus (Yellow catfish 172 calicivirus) challenge was performed using the poly(I:C) in this study.” I assume the poly(I:C) reagent was purchase from a company and is not the “Yellow catfish 172 calicivirus”. Please indicate in the methods the origin of the poly(I:C) used for the experiments and modify the sentence to “…and the virus-like challenge was performed using the poly(I:C) in this study.”

b) I think the sentence in the reply letter “The corresponding 5-10 upstream and downstream genes were selected for linear analysis to determine the location of TLR3. “ should be added to the main manuscript methods as no indication of the length of the genome regions analysed in the gene synteny map is provided. This will give the reader an idea of the relative distance between the genes that were mapped in the Figure.

Congratulations for the nice study.

Author Response

Thank you very much for your approval of our manuscript. We have revised our manuscript accordingly. Please refer to the PDF file for details.
